# Unsupervised Pre-training Improves Tooth Segmentation in 3-Dimensional Intraoral Mesh Scans

**Xiaoxuan He**[1,4]                                         Xiaoxuan_He@zju.edu.cn

**Hualiang Wang**[1]                                       hualiang_wang@zju.edu.cn

**Haoji Hu**[1]                                                        haoji_hu@zju.edu.cn

**Jianfei Yang**[2]                                               yang0478@e.ntu.edu.sg

**Yang Feng**[3]                                           fengyang@angelalign.com

**Gaoang Wang**[4]                                    gaoangwang@intl.zju.edu.cn

**Zuozhu Liu**[*1,4]                                         zuozhuliu@intl.zju.edu.cn

[1] *College of Information Science and Electronic Engineering, Zhejiang University*

[2] *School of Electrical and Electronic Engineering, Nanyang Technological University*

[3] *Angelalign Inc.*

[4] *ZJU-UIUC institute, ZJU-Angelaling R&D institute for Intelligent Healthcare Zhejiang University, Zhejiang, 314400, China*

**Editors:** Under Review for MIDL 2022

## Abstract

Accurate tooth segmentation in 3-Dimensional (3D) intraoral scanned (IOS) mesh data is an essential step for many practical dental applications. Recent research highlights the success of deep learning based methods for end-to-end 3D tooth segmentation, yet most of them are only trained or validated with a small dataset as annotating 3D IOS dental surfaces requires complex pipelines and intensive human efforts. In this paper, we propose a novel method to boost the performance of 3D tooth segmentation leveraging large-scale unlabeled IOS data. Our tooth segmentation network is first pre-trained with an unsupervised learning framework and point-wise contrastive learning loss on the large-scale unlabeled dataset and subsequently fine-tuned on a small labeled dataset. With the same amount of annotated samples, our method can achieve a mIoU of 89.38%, significantly outperforming the supervised counterpart. Moreover, our method can achieve better performance with only 40% of the annotated samples as compared to the fully supervised baselines. To the best of our knowledge, we present the first attempt of unsupervised pretraining for 3D tooth segmentation, demonstrating its strong potential in reducing human efforts for annotation and verification.

**Keywords:** Unsupervised Learning, Tooth Segmentation, 3D Intraoral Scan

## 1. Introduction

With the development of Computer-Aided Design (CAD) techniques, digital dentistry has attracted tremendous attention with various significant breakthroughs (Wu et al., 2014), (Zanjani et al., 2019a), (Zanjani et al., 2019b), (Sun et al., 2020), (Zhang et al., 2020), (Wu et al., 2021). In many dental diagnosis scenarios, such as orthodontics and implant, the first crucial step is to precisely recognize individual teeth and the gingiva in the 3-Dimensional (3D) intra-oral scanned (IOS) tooth data collected from patients (Yuan et al.,

---

*  Corresponding Author

2010). In practice, a single IOS mesh for the upper or low jaw usually consists of more than 100,000 triangular faces. It usually takes about 15 to 30 minutes for an experienced expert to manually or interactively annotate a half jaw, which is undoubtedly cumbersome and labor-intensive (Hao et al., 2021). To enable more efficient treatment planning, automated strategies are highly demanded for real-world clinical applications.

A lot of works have launched attempts to address the 3D tooth segmentation task in IOS meshes. Traditional geometry-based methods extract hand-crafted features such as curvatures from IOS meshes to design decision rules for segmentation (Kondo et al., 2004), (Li et al., 2007), (Kumar et al., 2011), (Fan et al., 2014), (Li and Wang, 2016). Recently, many deep learning based methods are proposed with superior performance. Some works first extract predefined features and subsequently apply the 2D or 3D convolutional neural networks for 3D tooth semantic segmentation (Tian et al., 2019). There are also methods which design specific neural network architectures for end-to-end tooth segmentation, such as MeshSegNet (Lian et al., 2020), DC-Nets (Hao et al., 2021), TSegNet (Cui et al., 2021), Mask-MCNet (Zanjani et al., 2019a). However, most of these methods are only trained or validated with a small dataset, e.g., less than 50 IOS meshes, as annotating 3D IOS dental surfaces requires complex pipelines and intensive human efforts. Moreover, when these methods are evaluated in clinical settings, their performance always degrades due to the inferior generalization ability across diverse anatomical tooth features. Though the DC-Net presents a clinical applicability test, the annotated dataset is not publicly available due to privacy issues. As a result, it's hard for the community to make further advancements to meet the requirements for clinical usages.

Recent research has witnessed the great success of unsupervised pre-training strategies for various computer vision (Chen et al., 2020), (Grill et al., 2020), (He et al., 2020) and natural language processing tasks (Devlin et al., 2018). As for 3D vision, several pioneering works also investigate unsupervised pretraining for 3D point cloud processing via occlusion completion, contrastive learning or spatio-temporal representation learning strategies (Wang et al., 2021), (Huang et al., 2021),(Xie et al., 2020). However, these methods are not originally designated for 3D tooth segmentation, while the IOS meshes usually contain more complicated topological features and heterogeneous anatomical geometry as compared to simple natural objects. Moreover, the IOS meshes are of very high resolution, while current voxel-based pretraining frameworks are unable to achieve satisfactory results with accurate fine-grained segmentation for clinical applications.

In this paper, we propose a novel method to boost the performance of 3D tooth segmentation with an unsupervised pre-training strategy that leverages large-scale unlabeled IOS meshes. We first construct a large 3D IOS mesh dataset, which consists of 6,000 unlabeled IOS meshes and 1,000 labeled meshes. To cope with the high-resolution and heterogeneous mesh data, we formulate the segmentation task over 3D dental mesh as a fine-grained point cloud semantic segmentation task, avoiding approximation errors in voxel-based methods. With the Dynamic Graph CNN (DGCNN) (Wang et al., 2019) as our backbone, our tooth segmentation network is first pre-trained with an unsupervised learning framework and point-level InfoNCE (Oord et al., 2018) loss. Afterward, the pre-trained backbone is modified to adapt to the downstream semantic segmentation task and fine-tuned on a small labeled dataset. Extensive experiments reveal that our method can achieve a mIoU of 89.38%, significantly outperforming the supervised counterparts when trained with the same

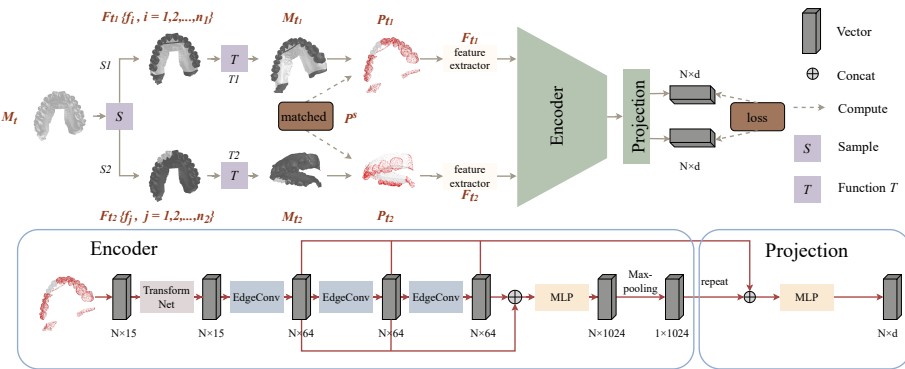

Figure 1: The proposed framework for unsupervised pre-training. Red points represent matched points, and dark faces represent $F_{t_1}$ and $F_{t_2}$. The output of the feature extractor is $h$ (consist of $h_c$, $h_n$, $h_s$) and used as input to the encoder.

amount of labeled samples. Moreover, our method can achieve segmentation performance on par with the fully supervised baselines with only 20-40% of the annotated samples. To the best of our knowledge, our work is the first attempt to employ unsupervised pre-training methods for 3D tooth segmentation, exhibiting strong potential for reducing human effort for annotation and verification.

## 2. Method

### 2.1. Overview

Given a 3D IOS mesh composed of many triangulated faces, 3D tooth segmentation aims to classify each face into different teeth and the gingiva following the Federation Dentaire Internationale (FDI) standard. Mathematically, for each face $f_i$ in the mesh, we want to annotate it with a label $y_i$, where $y_i \in \{0, 11-18, 21-28, 31-38, 41-48\}$ denotes the gingiva and FDI notations for the 32 permanent teeth, respectively. Our method includes two steps: unsupervised pre-training and supervised fine-tuning. In unsupervised pre-training, we first generate two augmented views of each unlabeled 3D IOS mesh and feed them into the segmentation backbone. A novel PointInfoNCE loss (Xie et al., 2020) function is adopted for unsupervised representation learning on a set of predefined matched points. Afterward, the pre-trained encoder is further fine-tuned in a supervised manner with a small number of labeled 3D IOS meshes.

### 2.2. Unsupervised pre-training

**Augmented data preprocessing.** It is not uncommon to generate asymmetric augmented input pairs for better representation learning in contrastive learning. As for 3D meshes or point clouds, the augmented input pair, which usually contains two augmentations with different views of the same input, should bring much more abundant and diverse training examples while discouraging the model from learning simple equivariance of the geometric transformation. Consequently, we generate two different views as our pre-training

input. Let $\mathcal{X} = \{M_t\}_{t=1}^{L}$ be the dataset with $L$ 3D IOS meshes where $M_t = (V, F)$ denotes the $t$-th sample with $V, F$ as mesh vertices and faces, respectively. The pipeline to generate asymmetric input pairs is elaborated as follows. Given $M_t$, we randomly sample two sets of faces $F_{t1} = \{f_i, i = 1, 2, .., n_1\}$ and $F_{t2} = \{f_j, j = 1, 2, ..., n_2\}$, where $n_1, n_2 > 10,000$ denotes the number of sampled faces, as illustrated in Figure 1. Such sampling process must ensure that there is a guaranteed overlap between the two sets to build a point-to-point correspondence, i.e., a set of matched points which plays a pivot role in subsequent pretraining, in the overlapping region over faces. Afterward, we apply two different transformations to $M_t$ to obtain two augmented views of it in two local coordinate systems, i.e., $M_{t1} = \mathcal{T}_1(M_t)$ and $M_{t2} = \mathcal{T}_2(M_t)$, where $\mathcal{T}_1, \mathcal{T}_2$ are different transformation augmentations as described below. The corresponding faces of $F_{t1}$ and $F_{t2}$ in $M_{t1}$ and $M_{t2}$ are extracted, and transformed to two point clouds $P_{t1}$ and $P_{t2}$, each with 10,000 points based on a uniform downsampling strategy (Hou et al., 2021),(Hao et al., 2021) over face centers. Finally, the correspondence mapping between points from the two point clouds are computed as $P^s = \{(i, j)\} = \Phi(P_{t1}, P_{t2})$, where $i$ and $j$ are the index of the matched points $x_i \in \mathbb{R}^3$ in $P_{t1}$ and $y_j \in \mathbb{R}^3$ in $P_{t2}$, and $\Phi$ defines a point-to-point mapping function in the world coordinate. The pipeline is illustrated in Figure 1.

**Transformation.** We apply different transformations on the 3D meshes to generate different augmented views. Mathematically, we define the transformation $\mathcal{T} = [R|t|S]$, in which $R \in SO(3)$ (3D rotation group in geometry) denotes the rotation, $t \in \mathbb{R}^3$ denotes translation, and $S$ denotes scaling operations, respectively. For the rotation $R$, we rotate the mesh with random angles (0 to 360°) around an arbitrary axis. Meanwhile, the function $t$ is devised to translate $M_t$ globally in the coordinates. The random scale function $S$ is designed to scale the $M_t$ with a factor randomly chosen from the range $[0.9, 1.1]$.

**Feature extraction.** After augmented data preprocessing, we generate two different point clouds $P_{t1}$ and $P_{t2}$. The 3D coordinate of each point is the center of the corresponding face, which is denoted as $h_c = [x_0, y_0, z_0] \in \mathbb{R}^3$. We further extract more geometrical features from the original mesh for each point. In particular, we compute the normal vector $h_n \in \mathbb{R}^3$, and a face shape descriptor $h_s \in \mathbb{R}^9$ as suggested in (Hao et al., 2021) with detailed computational steps attached in the Appendix. Finally, we concatenate the three features together as the feature vector of each point in our point cloud, leading to a 15-dimensional feature vector $h = \text{Concat}(h_c, h_n, h_s) \in \mathbb{R}^{15}$.

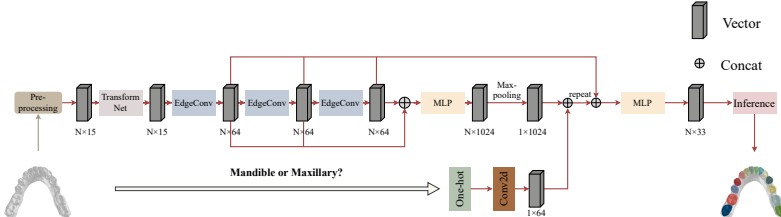

Figure 2: The architecture of supervised fine-tuning. $N = 20,000$ is the number of points. Similarly, the input of our network is $h$. The output of our encoder is a $1 \times 1024$ dimension global feature. Then we concat it with our one-hot categorical vector to construct a new global feature containing the prior knowledge.

**Unsupervised pre-training framework.** As shown in Figure 1, our unsupervised pre-training framework employs a standard contrastive learning architecture, which enables the encoder to learn the point-wise consistent representations by shrinking the distance between samples from the same asymmetric pair in the hidden space. Specifically, our framework includes two parts: a representation learning module and an InfoNCE loss function.

We design the encoder to learn feature representations of the extracted point clouds from 3D tooth data, as shown in Figure 1. Our encoder is inspired by the Dynamic Graph CNN (DGCNN) (Wang et al., 2019) with modifications to adapt to the 3D IOS data, which is of much higher resolution and morphological complexity. Let's consider $P_{t_1}$ only. It is firstly transformed into a standard feature space with the Transform Net (Qi et al., 2017a). Second, it is fed to three consecutive Edge-Conv blocks (Wang et al., 2019), which consist of a feature extractor based on k-nearest neighbor(kNN) strategy, three 2D convolutional layers, and a max-pooling aggregation operation. Based on an explicit local graph among neighborhood points defined by kNN, the Edge-Conv block updates the edge features with convolutional operations. The features used for kNN are the corresponding output from the previous block, leading to updated proximity defined on different hidden representations. Hence, the stacked Edge-Conv blocks can learn local features in the bottom layers and global semantic features in the top layers. With the concatenated representation from different layers, our backbone is able to capture both local topological geometry and global features for every point in $P_{t_1}$. Such representations are projected to a consistent hidden space with a projection head (i.e., a Multilayer Perceptron) for subsequent contrastive representation learning, following the standard conventions in many contrastive learning paradigms.

**Loss Function.** The InfoNCE loss (Oord et al., 2018) is proposed and has been widely used for unsupervised pre-training in 2D vision tasks. It is adopted by contrastive learning frameworks to conduct a dictionary query process. Here we define the PointInfoNCE loss (Xie et al., 2020) over points in the two augmented point clouds. We define the matched point pairs $(i, j)$ in $P^s$ as positive pairs, whose features $h_i$ and $h_j$ are obtained via the encoder and projection head. We further define $(i, k)$ as negative pairs if $\exists(\cdot, k) \in P^s, k \neq j$. In this case, we are considering points that have at least one matched point pairs in $P^s$ as the negative samples, ignoring all other non-match points for more efficient loss computation. Given the positive and negative pairs, the contrastive learning loss is defined as follows:

$$L = -\frac{1}{|P|} \sum_{(i,j)\in P} \log \frac{exp(h_i \cdot h_j/\tau)}{\sum_{(\cdot,k)\in P} exp(h_i \cdot h_k/\tau)}. \tag{1}$$

Optimizing over this PointInfoNCE loss function would minimize the distance between positive point pairs while maximizing the distance between negative point pairs, leading to good point-wise representations for further tooth semantic segmentation.

### 2.3. Supervised fine-tuning

The unsupervised pre-trained backbone is further modified and fine-tuned in a supervised manner for the downstream 3D tooth segmentation task. In particular, we use a one-hot categorical vector to denote the maxillary and mandible for the input half jaws, which is prior knowledge to avoid confusion about them during inference. The one-hot vector is further embedded with a convolutional layer and concatenated with the point-wise representations

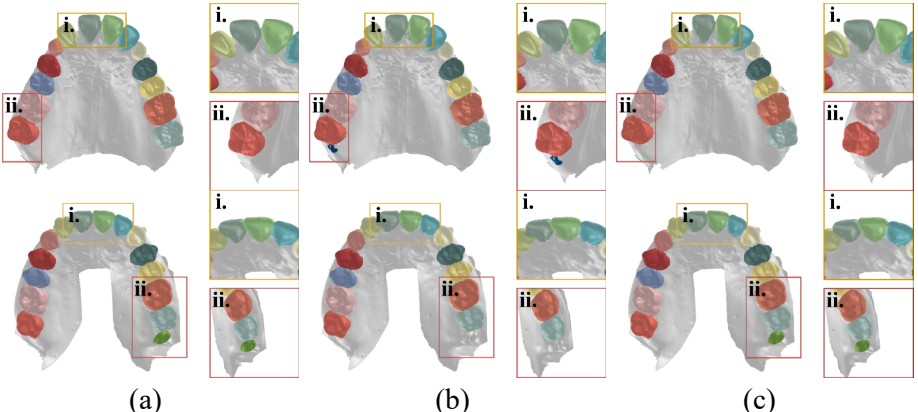

Figure 3: Visualization of DGCNN and our method for 3D Tooth Segmentation. Each row for a case. (a) The ground truth; (b) DGCNN; (c) Ours

from the pre-trained backbone. The fused representation is used for semantic segmentation over 32 permanent teeth and the gingiva with a multilayer perceptron composed of two fully-connected layers and a dropout layer with a keep probability of 0.4. We use the cross-entropy loss for supervised fine-tuning. During fine-tuning, the pre-trained weights serve as initial weights for the supervised backbone, leading to much faster convergence and better performance as shown in experiments. As for inference, we can not feed all the points in IOS meshes (e.g., 100,000+ points) to our network due to overloaded GPU memory, while performing multi-step inference for each of the 10,000 points is quite time-consuming as well, e.g., we need 10 inference steps for 100,000 points. In this work, we only inference 40,000 randomly sampled points for each mesh, and use a simple kNN based voting mechanism to generate semantic labels for all the rest points. Such a strategy maintains roughly the same performance as the multi-step method but with better efficiency.

## 3. Experiment

### 3.1. Implement Details

**Dataset.** We collect a large 3D IOS tooth mesh dataset, which consists of 6,000 unlabeled and 1,000 labeled 3D IOS mesh data. We split the labeled data to 60% for training, 20% for validation and 20% for testing. More details about experimental setup are in the Appendix.

### 3.2. 3D Tooth Segmentation Performance

We compare our method with extensive baselines in recent works (e.g., PointNet (Qi et al., 2017a), PointNet++ (Qi et al., 2017b), DGCNN (Wang et al., 2019), MeshSegNet (Lian et al., 2020) and DC-Net (Hao et al., 2021)). The results are reported in Table 1. We can notice that our method can surpass DGCNN with a significant improvement of 2.29% mIoU. This demonstrates that our unsupervised pre-training method can achieve non-trivial performance improvement, even with the same backbone as its supervised counterparts.

Table 1: Segmentation performance of our method and supervised baselines.

| Method | Mandible | | | Maxillary | | | All | | |
|---|---|---|---|---|---|---|---|---|---|
| | Acc | mIoU | DSC | Acc | mIoU | DSC | Acc | mIoU | DSC |
| PointNet | 72.44 | 67.25 | 75.09 | 75.66 | 72.92 | 79.97 | 74.29 | 70.51 | 77.89 |
| PointNet++ | 68.52 | 62.50 | 71.34 | 70.41 | 71.23 | 78.59 | 69.61 | 67.52 | 75.51 |
| MeshSegNet | 90.88 | 77.15 | 82.63 | 93.33 | 79.36 | 84.33 | 92.29 | 78.42 | 83.61 |
| DGCNN | 94.97 | 83.80 | 87.61 | 96.71 | 89.51 | 92.50 | 95.97 | 87.09 | 90.42 |
| DC-Net | 93.81 | 86.94 | 89.74 | 96.10 | 90.89 | 92.88 | 95.12 | 89.21 | 91.54 |
| **Ours** | **96.05** | **87.18** | **90.37** | **97.13** | **91.00** | **93.79** | **96.67** | **89.38** | **92.33** |

Table 2: Segmentation performance of pre-trained with different amount of data.

| Data Ratio | Mandible | | | Maxillary | | | All | | |
|---|---|---|---|---|---|---|---|---|---|
| | Acc | mIoU | DSC | Acc | mIoU | DSC | Acc | mIoU | DSC |
| 10% | 95.32 | 85.30 | 88.57 | 97.07 | 90.75 | 93.54 | 96.33(+0.36) | 88.43(+1.34) | 91.43(+1.01) |
| 50% | 95.59 | 86.22 | 89.43 | 97.09 | 90.39 | 93.17 | 96.45(+0.48) | 88.62(+1.53) | 91.58(+1.16) |
| 100% | 96.05 | 87.18 | 90.37 | 97.13 | 91.00 | 93.79 | 96.67(+0.7) | 89.38(+2.29) | 92.33(+1.91) |

DC-Net not only introduces a novel network, but also explores the improvement of inference. Different from our knn strategy, DC-Net optimizes the inference with graph cut algorithms, which significantly boosts the inference performance. However, even using knn, our method still exceeding the DC-Net with 0.79% DSC and 1.55% accuracy.

We further investigate the effect of pre-training with different amount of unlabeled data, with results in Table 2. When we use 10% of unlabeled 3D IOS data during pre-training, we still achieve 96.33% accuracy, 88.43% mIoU and 91.43% DSC. Compared to the supervised DGCNN model, it has 1.34% mIoU improvement, revealing that even using little unlabeled data, unsupervised pre-training can still achieve impressive performance. With the increasing amount of unlabeled pre-training data, all the evaluation metrics can be constantly improved, convincingly demonstrating the effectiveness of our method.

Table 3: Segmentation performance with limited labeled training data for fine-tuning

| Data Ratio | Train Strategy | Mandible | | | Maxillary | | | All | | |
|---|---|---|---|---|---|---|---|---|---|---|
| | | Acc | mIoU | DSC | Acc | mIoU | DSC | Acc | mIoU | DSC |
| 1% | from scratch | 63.00 | 39.06 | 48.02 | 73.83 | 51.22 | 60.32 | 69.23 | 46.05 | 55.09 |
| | Our | 80.75 | 55.81 | 63.98 | 87.55 | 67.45 | 74.70 | 84.66 | 62.50 | 70.14 |
| 5% | from scratch | 88.82 | 68.86 | 75.16 | 91.22 | 74.07 | 79.25 | 90.20 | 71.86 | 77.51 |
| | Our | 89.08 | 69.45 | 74.91 | 92.90 | 78.07 | 82.39 | 91.28 | 74.41 | 79.21 |
| 10% | from scratch | 90.75 | 73.86 | 79.62 | 93.03 | 78.22 | 82.81 | 92.06 | 76.37 | 81.46 |
| | Our | 91.99 | 75.41 | 80.40 | 94.40 | 82.47 | 86.40 | 93.37 | 79.47 | 83.85 |
| 20% | from scratch | 91.95 | 76.12 | 81.07 | 94.22 | 80.50 | 84.57 | 93.26 | 78.64 | 83.08 |
| | Our | 93.97 | 81.11 | 85.14 | 95.06 | 85.22 | 88.82 | 94.60 | 83.48 | 87.25 |
| 40% | from scratch | 94.19 | 82.65 | 86.62 | 96.16 | 86.93 | 90.21 | 95.32 | 85.11 | 88.68 |
| | Our | 94.89 | 84.17 | 87.67 | 96.81 | 89.74 | 92.71 | 95.99 | 87.37 | 90.57 |
| 100% | from scratch | 94.97 | 83.80 | 87.61 | 96.71 | 89.51 | 92.50 | 95.97 | 87.09 | 90.42 |
| | Our | **96.05** | **87.18** | **90.37** | **97.13** | **91.00** | **93.79** | **96.67** | **89.38** | **92.33** |

We also conduct a series of experiments to evaluate the effectiveness of using different amounts, i.e., 1%, 5%, 10%, 20%, 40%, 100%, of the labeled data during fine-tuning, with

Table 4: The result on 3D IOS mesh dataset of different transformations

| Transformations | Mandible | | | Maxillary | | | All | | |
|---|---|---|---|---|---|---|---|---|---|
| | Acc | mIoU | DSC | Acc | mIoU | DSC | Acc | mIoU | DSC |
| translation | 95.38 | 84.30 | 87.84 | 96.70 | 89.41 | 92.36 | 96.14(+0.17) | 87.24(+0.15) | 90.44(+0.02) |
| rotation | 95.70 | 86.10 | 89.39 | 97.07 | 90.81 | 93.68 | 96.48(+0.51) | 88.81(+1.72) | 91.86(+1.44) |
| scale | 95.52 | 83.85 | 87.43 | 96.74 | 90.19 | 93.16 | 96.22(+0.25) | 87.50(+0.41) | 90.72(+0.3) |
| translation + rotation | **96.05** | **87.18** | **90.37** | **97.13** | **91.00** | **93.79** | **96.67(+0.70)** | **89.38(+2.29)** | **92.33(+1.91)** |

results shown in Table 3. When trained with only 1% labeled data, DGCNN trained without weight initialization from pre-trained models can only achieve 69.23% accuracy, 46.05% mIoU, and 55.09% DSC. In stark contrast, our method significantly outperforms it with 84.66% accuracy, 62.50 % mIoU, and 70.14% DSC. The above two experiments demonstrate that unsupervised pre-training is an effective solution for tooth segmentation when the annotated data is severely limited. We can also notice that, with 40% labeled data, our method surprisingly achieves 87.37% mIoU, even surpassing the supervised DGCNN model trained with 100% labeled data by 0.28% mIoU. Meanwhile, when 100% labeled data is available, our method further extends the superiority to 2.29% mIoU.

### 3.3. Ablation Studies

Augmentation strategies usually have a non-trivial influence over the performance of un-supervised pretraining methods. Hence, we conduct an ablation study to quantitatively evaluate the effect of different augmentation methods during pretraining. The results are shown in Table 4. We can notice that rotation brings more remarkable improvement compared to translation and scale operations for unsupervised pre-training. When translation and rotation operations are simultaneously adopted, satisfactory performance is achieved.

### 3.4. Visualization

We visualize two segmentations to demonstrate the superiority of our method. In the first case, DGCNN makes mistakes between the incisors, and also wrongly treats part of the gums as third molars. In the second case, DGCNN makes a severe mistake by failing to identify a small third molar. In stark contrast, our method is able to correctly recognize such tiny teeth. There are certainly some limitations of our method as well, which can be found in the Appendix and addressed in the future.

### 4. Conclusion

In this paper, we propose a novel unsupervised pre-training strategy which helps significantly boost the performance of 3D tooth segmentation. Our method achieves 2.29% mIoU improvement compared to the supervised counterparts and is on par with the supervised method with only 20-40% labeled data. The extensive experiments convincingly corroborate the effectiveness of the proposed unsupervised pre-training strategy for helping alleviate the necessity of large-scale labeled training data for accurate 3D tooth segmentation. We expect our future work for improved performance over complicated IOS scans with heterogeneous anatomical features for clinically applicable diagnosis.

## Acknowledgements

This work was supported by the National Natural Science Foundation of China (U21B2004 and 62106222), and Zhejiang Provincial key RD Program of China (2021C01119).

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

## Appendix A. Experiment Setup

In unsupervised pre-training period, we use the Adam optimizer with learning rate $\eta_1 = 0.001$ and a exponentially decay factor 0.99. The hyperparameter $\tau = 0.4$ in the PointInfoNCE loss. During fine-tuning, we use SGD with initial learning rate $\eta_2 = 0.1$ that decays until 0.001 with cosine annealing. We use $k = 25$ for the kNN step in Edge-Conv blocks, and the network is trained with a batch size $bs = 4$ over $N = 10,000$ points in pre-training and $N = 20,000$ points in fine-tuning. Our code will be released for a better understanding. **Metrics.** We comprehensively evaluate the performance of our method with various metrics, i.e., mIoU, Dice Similarity Coefficient(DSC), point-level classification accuracy, precision, and recall. For one class $l$, we denote the sets of prediction and ground truth as $P_l$ and $T_l$. The Dice Similariy Coefficient(DSC) is usually used to measure similarity of two sets. For a class $l$, the DSC is denoted as $DSC = \frac{2|P_l \cap T_l|}{|P_l| + |T_l|}$, where $l = \{0, \text{11-18, 21-28, 31-38,} \\ \text{41-48}\}$.

## Appendix B. Feature Vector

Now we elaborate on how to define the shape of a face. For each face, we have three vertices $v_i = [x_i, y_i, z_i]_{i=1}^3$ and a face center $h_c = [x_0, y_0, z_0]$, so we can use the relative position relationship between $\{v_i\}_{i=1}^3$ and $h_c$ to define face shape descriptor, as $h_s = \text{Concat}([v_i - h_c]_{i=1}^3)$. ($\text{Concat}(\cdot)$ represents concatenate operation for vectors). Finally, we connect the above features together as the feature vector of each point in our point cloud, as $h = \text{Concat}(h_c, h_n, h_s) \in \mathbb{R}^{15}$, a 15-dimensional feature vector.

## Appendix C. Backbone

**Projection Head.** As illustrated in Figure 1, the overall architecture of our unsupervised backbone consists of an encoder and a projection head. The structure of the encoder has been elaborated on before. Now we describe the composition of the projection head in detail. The projection head, composed of a set of cascaded multilayer perceptron (i.e., 512,256,32), projects the output of our encoder to a consistent hidden space. We refer to (Xie et al., 2020) to set the number of output neurons of our projection head as 32.
**Supervised fine-tuning.** Overall, the entire deep learning architecture corresponding to Figure 2 is composed of: Input $\rightarrow$ Transform Net $\rightarrow$ EdgeConv $\rightarrow$ EdgeConv $\rightarrow$ EdgeConv $\rightarrow$ Conv2D[1024] $\rightarrow$ maxpool $\rightarrow$ Conv2D[256] $\rightarrow$ Dropout $\rightarrow$ Conv2D[256] $\rightarrow$ Dropout $\rightarrow$ Conv2D[128] $\rightarrow$ Output. Meanwhile, for our categorical vector, we first cope it with a Conv2D[64] layer and then feed the output into our backbone. Based on the above architecture, the network is capable of handling 3D teeth data with much higher resolution and morphological complexity.

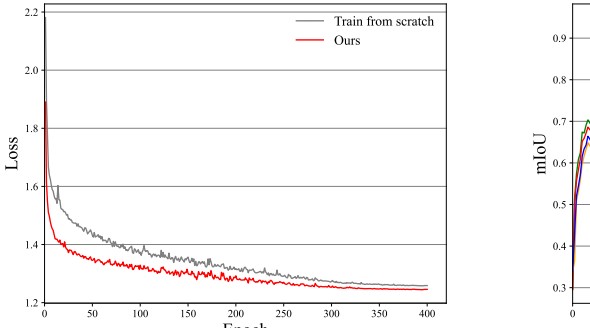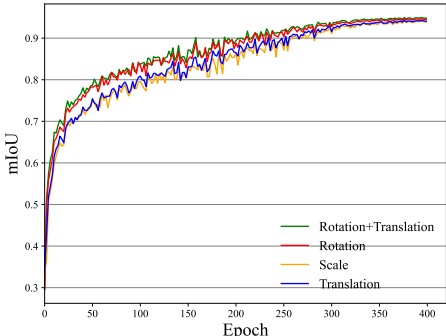

Figure 4: The curve of our training step in supervised fine-tuning. The left figure shows the curve of loss between train from scratch and ours; The right figure shows the curves of different transformations.

## Appendix D. More experiments

**Segmentation results for each category.**

Table 5: Mandible

| Mandible | mIoU | Precision | Dice | Recall |
|---|---|---|---|---|
| 0 | 95.24 | 98.14 | 97.55 | 96.98 |
| 31 | 83.89 | 90.38 | 88.73 | 91.66 |
| 32 | 87.39 | 93.25 | 91.28 | 92.21 |
| 33 | 90.90 | 93.84 | 94.25 | 94.95 |
| 34 | 85.36 | 94.23 | 88.16 | 89.01 |
| 35 | 90.60 | 94.47 | 93.91 | 94.80 |
| 36 | 91.62 | 94.98 | 94.50 | 95.54 |
| 37 | 90.11 | 92.83 | 92.79 | 94.18 |
| 38 | **75.44** | **95.00** | **77.54** | **80.30** |
| 41 | 85.83 | 92.10 | 90.33 | 91.34 |
| 42 | 86.77 | 92.37 | 90.75 | 93.00 |
| 43 | 91.99 | 95.60 | 95.16 | 94.97 |
| 44 | 85.10 | 95.21 | 87.63 | 88.60 |
| 45 | 89.65 | 94.49 | 92.99 | 94.22 |
| 46 | 89.66 | 94.07 | 92.53 | 93.89 |
| 47 | 89.21 | 92.49 | 92.17 | 92.30 |
| 48 | **73.30** | **92.99** | **75.94** | **79.77** |

Table 6: Maxillary

| Maxillary | mIoU | Precision | Dice | Recall |
|---|---|---|---|---|
| 0 | 95.94 | 98.05 | 97.92 | 97.81 |
| 11 | 91.82 | 94.97 | 95.07 | 95.27 |
| 12 | 90.31 | 95.99 | 93.63 | 94.07 |
| 13 | 91.74 | 94.53 | 94.86 | 96.61 |
| 14 | 89.90 | 95.54 | 92.56 | 94.25 |
| 15 | 92.73 | 96.25 | 95.58 | 96.25 |
| 16 | 94.09 | 96.13 | 96.54 | 97.40 |
| 17 | 91.88 | 95.48 | 94.92 | 95.18 |
| 18 | **85.51** | **97.39** | **88.08** | **87.72** |
| 21 | 92.10 | 94.83 | 95.27 | 95.85 |
| 22 | 90.67 | 94.25 | 94.02 | 95.35 |
| 23 | 92.63 | 95.61 | 95.43 | 96.16 |
| 24 | 86.85 | 95.11 | 89.29 | 91.23 |
| 25 | 90.42 | 96.04 | 93.61 | 94.02 |
| 26 | 95.20 | 97.15 | 97.48 | 97.94 |
| 27 | 91.73 | 95.75 | 94.48 | 94.29 |
| 28 | **83.50** | **95.55** | **85.61** | **87.35** |

**Curves.** The left part of Figure 4 plots the curve of loss value during the fine-tuning period. Gray and red lines represent the loss of each epoch with training from scratch and fine-tuning strategies, respectively, using 100% labeled data. Compared to training from scratch, using fine-tuning after unsupervised pre-training can achieve faster convergence and lower loss values in the whole training period. The right part of Figure 4 demonstrates curves of mIoU per epoch during training under different transformations.

Due to matched pairs different between our data, so we need to sample them to specify the size to compute loss. Table 7 shows the impact of the number of matched pairs in $P^s$. Using 4096 pairs can achieve the best performance, compared to others.

Table 7: The results on 3D IOS mesh dataset of different size of $P^s$

| Number of faces | Mandible | | | Maxillary | | | All | | |
|---|---|---|---|---|---|---|---|---|---|
| | Acc | mIoU | DSC | Acc | mIoU | DSC | Acc | mIoU | DSC |
| 1024 | 95.69 | 85.69 | 89.07 | 97.04 | 90.94 | 93.77 | 96.46 | 88.71 | 91.77 |
| 2048 | 95.27 | 85.47 | 88.85 | 96.99 | 90.84 | 93.69 | 96.26 | 88.56 | 91.63 |
| 4096 | **96.05** | **87.18** | **90.37** | **97.13** | **91.00** | **93.79** | **96.67** | **89.38** | **92.33** |

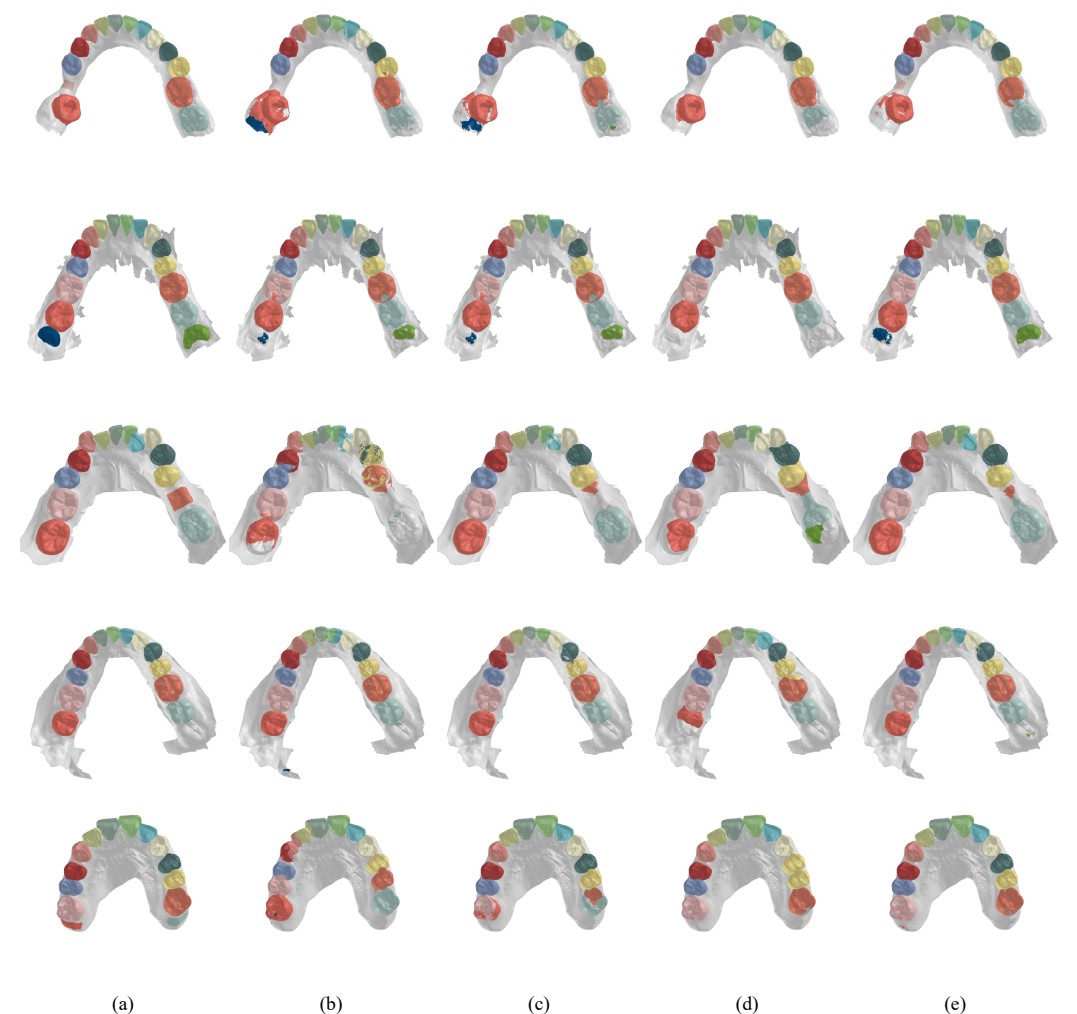

(a)   (b)   (c)   (d)   (e)

Figure 5: Visualization of several methods for 3D Tooth Segmentation. Each row for a case. (a) The ground truth; (b) MeshSegNet; (c) DGCNN; (d) DC-Net; (e) Ours

## Appendix E. More Segmentation Results

Figure 5 shows more results of tooth segmentation. Our method outperforms the state-of-the-art methods for 3D IOS mesh segmentation. Different from DC-Net, which optimizing

the inference with graph cut algorithms, our method aims at improve the performance of the tooth segmentation network with unsupervised pre-training. The visualization demonstrate that our method has superior in identifying 18, 28, 38, 48 teeth than other methods.

## Appendix F. Pre-training on other backbone

Figure 6: The curve of our training step in supervised fine-tuning

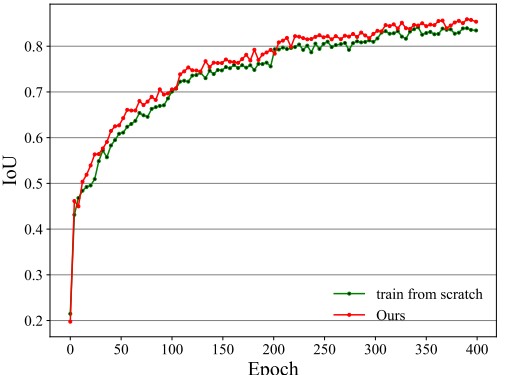

Table 8: Segmentation performance of our method on PointNet

| Methods | Mandible | Maxillary | All |
|---|---|---|---|
| | DSC | DSC | DSC |
| train from scratch | 75.09 | 79.97 | 77.89 |
| Ours | 78.08 | 82.42 | 80.58 |

We also apply our pre-training method to other backbone. Figure 6 shows the curve of mIoU on training set between training from scratch and ours based on PointNet (Qi et al., 2017a). Table 8 demonstrate the DSC of our method and training from scratch. Meanwhile, We also tested other metrics(e.g. accuracy, mIoU). Our pre-training method achieves 73.99% mIoU and 80.58% DSC, exceeding the result of training from scratch with 2.69% DSC and 3.48% mIoU.

