# OpenReview forum: "Unsupervised Pre-training Improves Tooth Segmentation in 3-Dimensional Intraoral Mesh Scans"
_MIDL.io/2022/Conference — MIDL 2022_

### Official Review · Reviewer_JN71 · 2022-01-24

**Confidence:** 4
**Preliminary Rating:** 4
**Recommendation:** Poster

**Summary:**

The authors propose to use a pre-training strategy using self-supervised loss to learn good network initialization with large unlabeled data that can be later fine-tuned with limited labels for 3D tooth segmentation. A large 3D IOS mesh data with unlabeled and labeled meshes was constructed and used for evaluation. With a similar number of annotations, the proposed approach yields higher segmentation performance than supervised baselines. It also requires only 40\% of annotations to match the fully supervised baseline.


**Strengths:**

1. It is a well-written paper, and the earlier works and their shortcomings are well explained.
2. The methods section is clear and appropriate details such as transformations and loss functions definitions in pre-training are explained.
3. The construction of the dataset is neat. The training and hyper-parameter details are clearly mentioned.
4. The experiments presented with different amounts of labeled and unlabeled data are useful in understanding where the major benefits for the proposed approach are obtained.
5. Also, the qualitative visualization of the results is decent.


**Weaknesses:**

P1. The authors could benefit from a better caption for Figure 1. They can provide more details about the method and explain the notations used in the caption or in the figure.

P2. Currently, the concatenated feature used to define the Point InfoNCE loss as in Eqn. 1 is used for pre-training is not clearly described in Figure 1. The concatenated feature consists of h_c (feature from the encoder-projection head network), h_n (normal vector), and h_s (face shape descriptor), while in Figure only h_c is input for the loss computation.

P3. In Figure 2, for the fine-tuning, it is not clear if the 1x64 feature vector from categorical information is concatenated with the feature vector h = (h_c,h_n,h_s) formed from 3 features or with only h_c feature output from the encoder-MLP network?

P4. The authors could benefit from comparing with semi-supervised approaches [1,2], data augmentation [3,4], pretext task based self-supervised learning methods [4,6,7] in the literature that work with limited annotations and unlabeled data that have shown to provide improvements over fully supervised baselines. This would make the article more complete by positioning it w.r.t to relevant works in the literature.
[1] Bai et al. Semi-supervised learning for network-based cardiac mr image segmentation.
[2] Xie el al. Self-training with Noisy Student improves ImageNet classification.
[3] Zhang et al. mixup: Beyond empirical risk minimization.
[4] Chaitanya et al. mixup: Beyond empirical risk minimization.
[5] Gidaris et al. Unsupervised representation learning by predicting image rotations.
[6] Pathak et al. Context encoders: Feature learning by inpainting.
[7] Chen et al. Self-supervised learning for medical image analysis using image context restoration

**Deanonymize Review:**

no

**Detailed Comments:**

The other minor comments are mentioned below:

1. It is interesting to observe that the DSC and mIoU for all different ratios (10\%, 50\%, 100\%) of unlabeled data are fairly similar (within 1\%) given in Table 2. Can the authors please update the corresponding results text that increasing the number of unlabeled data does not drastically affect the final performance?

2. Similarly, the effect of different augmentations seems to not affect the performance much (changes in metrics are within 1-2\%).

3. In Figure 3, there are two smaller zoomed-in areas referred to as (a), (b) in each of the ground truth and methods' results. Can they change this notation to (i, ii) or something different to (a, b) as they are already used to refer to ground truths and methods.


**Final Rating After The Rebuttal:**

4: Weak Accept

**Justification Of The Final Rating:**

The authors have addressed most of the concerns raised and have provided valid justifications or clarifications.
They have made the appropriate changes in the revised draft to include the suggested comments.
I am happy with the revised draft and have no further comments.

**Paper Type:**

methodological development

**Questions To Address In The Rebuttal:**

I would request the authors to address the points mentioned in the weakness section.

1. First, they can clarify the notations used in Figures 1 and 2 and the corresponding text and equations such that all of them match [P1,P2,P3].
2. Next, the authors can benefit from providing appropriate comparisons with works in the literature that use limited annotations and unlabeled data such as semi-supervised, data augmentation, and other self-supervised learning approaches [P4].

**Special Issue:**

no

---

### Official Review · Reviewer_Z3yE · 2022-01-24

**Confidence:** 4
**Preliminary Rating:** 3
**Recommendation:** Poster

**Summary:**

*Paper Typ The paper addresses tooth segmentation of intra-oral scans (IOS) in a semi-supervised setting. The authors adopt the concept of PointContrast for self-supervised pre-training, which is followed by supervised fine-tuning. As backbone, the authors use a modified Dynamic Graph CNN. Experiments show that self-supervised pre-training enables to surpass the baseline model by 2.3% points DSC or to match the baseline model’s performance at only 40% of the training data.


**Strengths:**

- This is the first work to explore semi-supervised / self-supervised learning in the context of IOS segmentation. And given the required effort to manually annotate a scan, methods to effectively exploit unlabeled data are highly relevant.
 - Quantitative comparison between the proposed method and the baseline shows good results.


**Weaknesses:**

- The self-supervised learning strategy appears identical to PointContrast by Xie et al. 2020. Authors should clarify if / how their method differs from this work.
 - In the experiments, the self-supervised method is only compared to the baseline model and not to stronger competitors. Given the semi-supervised setting, it would be interesting to compare to other methods like self-training, consistency regularization or a mixture, as discussed in [1].
 - The self-supervised training scheme is only validated for a single backbone (DGCNN). Most interesting would be its effect when combined with existing models that are specifically tailored for IOS segmentation. Maybe, the authors would provide code on request or the models can be reimplemented? Otherwise, the self-supervised scheme could be combined with PointNet and PointNet++ to check for similar performance gains as fir DGCNN.


**Deanonymize Review:**

no

**Detailed Comments:**

The spacing between figures / table and text is quite narrow such that pp. 4,5,7,8 are very crowded.
 - The first citation of He et al. 2020 on p. 2 seems to be misplaced.
 - The specified size of the labeled dataset is inconsistent (600 on p.2 and 1000 on p.6)
 - Which data augmentations are finally used for the best model? According to Sec 2.2, it is rotation, translation and scaling. According to Tab. 4, it is just translation and rotation.
- Typo in beginning of Sec. 3.2: “we” > “We”

[1] Jiang, Li, et al. "Guided Point Contrastive Learning for Semi-supervised Point Cloud Semantic Segmentation." ICCV. 2021.

**Final Rating After The Rebuttal:**

4: Weak Accept

**Justification Of The Final Rating:**

The authors partially addressed my concerns and added another backbone (PointNet) for which their self-supervised learning step leads to an improvement. I would strongly encourage the authors to share the view of open science and open data that defines the MIDL community and to release their source code (this was not directly clear or answered) as this is of high importance for reproducibility.

**Paper Type:**

both

**Questions To Address In The Rebuttal:**

Questions To Address In The Rebuttal: How does the method compare to stronger baselines (self-training, consistency regularization)?
- How effective is the method with different network architectures?


**Special Issue:**

no

---

### Official Review · Reviewer_SJbV · 2022-01-25

**Confidence:** 4
**Preliminary Rating:** 3
**Recommendation:** Poster

**Summary:**

This paper presents a new pre-training strategy for 3D tooth segmentation. Specifically, they collect a large-scale dataset and utilize point-wise contrastive learning loss to pre-train the whole network. They formulate the segmentation tasks as a fine-grained point cloud segmentation task and use DGCNN for segmentation. Experiments demonstrate the effectiveness of the proposed method and save annotation effort. It sets a strong baseline for this kind of application and would benefit the community if authors can make the dataset and code public.

**Strengths:**

1. The proposed framework is interesting.
2. The collected dataset is large and it should be beneficial in the clinic. And experiments demonstrate the effectiveness of the proposed method.
3. Overall, the writing is good.

**Weaknesses:**

1. Writing can be further improved, especially authors should stress the difference of semantic segmentation based on mesh data.
2. Although the codes of some baselines are not available, authors should contact the authors of those papers or re-implement these by themselves.


**Deanonymize Review:**

no

**Detailed Comments:**

1. Novelty
Authors should stress the difference of semantic segmentation based on mesh data. Further, they should point out what is designed to address those challenges related to this kind of application. The writing should be improved.

2. Experiments
2.1 Baselines
Authors should contact the authors of those baseline papers or re-implement these baselines by themselves.

2.2 Table 3. A figure would be better to present these results.

2.3 Table 2. Authors should analyze why utilizing 10% of data for pre-training can achieve such high performance.
Authors may consider reducing the amount of labeled data for fine-tuning to show the importance of the amount of unlabeled data for pre-training.




**Final Rating After The Rebuttal:**

4: Weak Accept

**Justification Of The Final Rating:**

The authors address most of my concerns, including clarifying the differences compared with previous works, adding necessary baselines and comparison experiments, modifying and editing contents. It would benefit the whole community if they can publish the code after getting accepted.

**Paper Type:**

both

**Questions To Address In The Rebuttal:**

Please see my detailed comments above, especially:

1. Writing
Authors should more clearly show the difference of this kind of application compared to conventional semantic segmentation (just like what is shown in Fig. 1, including mesh/point cloud, etc).

2. Missing baselines.

3. Implementation details, such as input data dimensions, intermediate feature dimensions, etc.

4. Add qualitative results of baseline methods in Fig. 3.



**Special Issue:**

no

---

### Meta-Review · Area_Chair_HYTX · 2022-02-19

**Recommendation:** Accept (Poster)
**Confidence:** 5

**Metareview:**

Despite the initial concerns from two reviewers, the authors rebuttal addressed most of the raised weaknesses/clarifications, which led to a final common score of weak accept among the three reviewers. After reading the reviewers comments and authors rebuttal, I side with the reviewers and recommend the acceptance (poster) of this work at MIDL'22.

---

### Decision · Program_Chairs · 2022-02-28

Accept